# Designed α-sheet peptides inhibit amyloid formation by targeting toxic oligomers

Gene Hopping[1], Jackson Kellock[1], Ravi Pratap Barnwal[2], Peter Law[1], James Bryers[1], Gabriele Varani[2], Byron Caughey[3], Valerie Daggett[1]*

[1]Department of Bioengineering, University of Washington, Seattle, United States; [2]Department of Chemistry, University of Washington, Seattle, United States; [3]Laboratory of Persistent Viral Diseases, Rocky Mountain Laboratories, National Institute of Allergy and Infectious Diseases, National Institutes of Health, Hamilton, United States

**Abstract** Previous studies suggest that the toxic soluble-oligomeric form of different amyloid proteins share a common backbone conformation, but the amorphous nature of this oligomer prevents its structural characterization by experiment. Based on molecular dynamics simulations we proposed that toxic intermediates of different amyloid proteins adopt a common, nonstandard secondary structure, called α-sheet. Here we report the experimental characterization of peptides designed to be complementary to the α-sheet conformation observed in the simulations. We demonstrate inhibition of aggregation in two different amyloid systems, β-amyloid peptide (Aβ) and transthyretin, by these designed α-sheet peptides. When immobilized the α-sheet designs preferentially bind species from solutions enriched in the toxic conformer compared with non-aggregated, nontoxic species or mature fibrils. The designs display characteristic spectroscopic signatures distinguishing them from conventional secondary structures, supporting α-sheet as a structure involved in the toxic oligomer stage of amyloid formation and paving the way for novel therapeutics and diagnostics.

*For correspondence: daggett@u.washington.edu

## Introduction

There are now over 40 different human amyloid diseases, each linked to the buildup of a specific precursor protein or peptide (*Chiti and Dobson, 2009*). These diseases involve the conversion of a protein from its soluble native state into insoluble amyloid fibrils, or, in the case of peptides, the conversion from a soluble, loosely structured form to fibrils. Given that many different sequences can form amyloid fibrils of similar architecture, there may be some common structural features of the prefibrillar amyloidogenic intermediates. X-ray fiber diffraction indicates that the insoluble, mature amyloid fibrils are composed of cross β-sheet structure (*Jahn et al., 2010*; *Eisenberg and Jucker, 2012*). Therefore, it is widely held that the formation of amyloid fibrils involves a transition to β-sheet structure in the amyloidogenic intermediate. However, the mechanism of self-assembly at the atomic level remains elusive. Another feature of these diseases is that soluble oligomeric intermediates, not the insoluble well-ordered fibrils, are preferentially responsible for cellular toxicity (*Bucciantini et al., 2002*; *Hardy and Selkoe, 2002*; *Tomic et al., 2009*; *Xue et al., 2009*). Similarly, the soluble oligomeric forms of the prion protein are the most infectious per unit protein (*Silveira et al., 2005*). As such, fibrils may be protective, at least up to a point, as their breakdown to smaller aggregates yields greater toxicity and infectivity. The discovery of a compound that promotes inclusion formation while reducing toxicity and cellular pathology supports this hypothesis (*Bodner et al., 2006*). In a similar vein, *Jiang et al. (2013)* demonstrated that binding the fibrillar state of Aβ could reduce toxicity presumably by shifting the equilibrium from oligomer to fibril.

**eLife digest** The build up of very thin fibres called amyloid fibrils is known to lead to more than 40 different human diseases, including Parkinson's disease and rheumatoid arthritis. These diseases involve soluble proteins or peptides joining other proteins or peptides to form the fibrils, which are not soluble. However, the damage is done by the time the fibrils form because soluble intermediate structures formed by the proteins and peptides are toxic. The development of methods that can detect these toxic intermediate structures could lead to earlier interventions before significant damage.

Amyloid fibrils are known to have a beta-sheet structure that is found in many protein systems. In 2004, based on computer simulations, researchers predicted that proteins and peptides that go on to form amyloid fibrils would pass through a related but less stable structure called an alpha-sheet, and that this structure would be toxic. Now Hopping et al., including some of the researchers involved in the 2004 work, have confirmed that the alpha-sheet structure is indeed involved in the formation of amyloid fibrils.

To do this Hopping et al. designed peptides with alpha-sheet structures that could bind to the alpha-sheet structures predicted by their simulations. When these complementary designed peptides were added to a solution of peptide that causes Alzheimer's Disease, or a protein that causes systemic amyloid disease, the designed peptides bound the toxic peptides or proteins and prevented the formation of fibrils.

The results of Hopping et al. suggest that designed alpha-sheet compounds might be able to capture peptides and proteins that are implicated in a wide variety of amyloid diseases, independent of their composition and native structure, by targeting the intermediate alpha-sheet structure. Future challenges include showing that most proteins and peptides pass through this intermediate structure as they form fibrils, and improving the sensitivity of the binding in the hope of developing diagnostics for amyloid diseases.

Research on the soluble oligomers has become critically important since there is a consensus that the soluble oligomer species is more toxic than mature fibrils (*Chiti and Dobson, 2006*; *Tomic et al., 2009*; *Xue et al., 2009*) and, while nontoxic, the fibrils are a reservoir for toxic oligomeric species (*Shahnawaz and Soto, 2012*). In fact, structural similarities between soluble oligomers from a range of unrelated proteins/peptides has been demonstrated by generation of an antibody that recognizes a common backbone structure (*Kayed et al., 2003*; *Tomic et al., 2009*). Glabe and co-workers developed an antibody (denoted A11) that is specific for soluble oligomeric intermediates derived from a variety of peptides and proteins, including Aβ(1-42), α-synuclein, islet amyloid polypeptide, polyglutamine, lysozyme, human insulin, and a prion peptide (*Glabe and Kayed, 2006*). The antibody does not, however, bind the corresponding insoluble fibrils (cross-β structure) or the natively folded precursors (various structures). Based on the specificity of the antibody for soluble oligomers with various sequences, it was proposed that the antibody might recognize a unique conformation of the backbone. This antibody inhibits toxicity associated with the intermediates, implying a common mechanism of toxicity and offering hope for a broad-based therapeutic agent.

Some years ago we 'discovered' a novel secondary structure, which we call '*α-sheet*', that is populated during molecular dynamics (MD) simulations of a range of amyloid proteins (and peptides) with different structures and sequences under amyloidogenic conditions (*Figure 1A*) (*Armen et al., 2004a*, *2004b*, *2005*; *Steward et al., 2008*). The position where the α-sheet forms along the sequence coincides with the most amyloidogenic regions of the proteins, as determined experimentally (*Armen et al., 2004a*). Consequently, we proposed that α-sheet is a common structure involved in the early stages of protein aggregation (*Armen et al., 2004a*; *Daggett, 2006*). In the course of characterizing the structure observed by MD, we learned that α-sheet was first predicted by Pauling and Corey and called '*polar pleated sheet*'. However, they ruled the structure energetically unfavorable and concluded, correctly, that the β-sheet structure would be favored in normal proteins (*Pauling and Corey, 1951*). An α-sheet is similar to a β-sheet, but instead of alternating main chain NH and C = O groups along the strands, an α-sheet has the NH groups aligned on one side and the carbonyls on the other. As such, the α-sheet has a molecular dipole and a very different hydrogen-bonding pattern across the sheet compared to a β-sheet. Interestingly, the main chain (Φ, Ψ) dihedral angles of the α-sheet

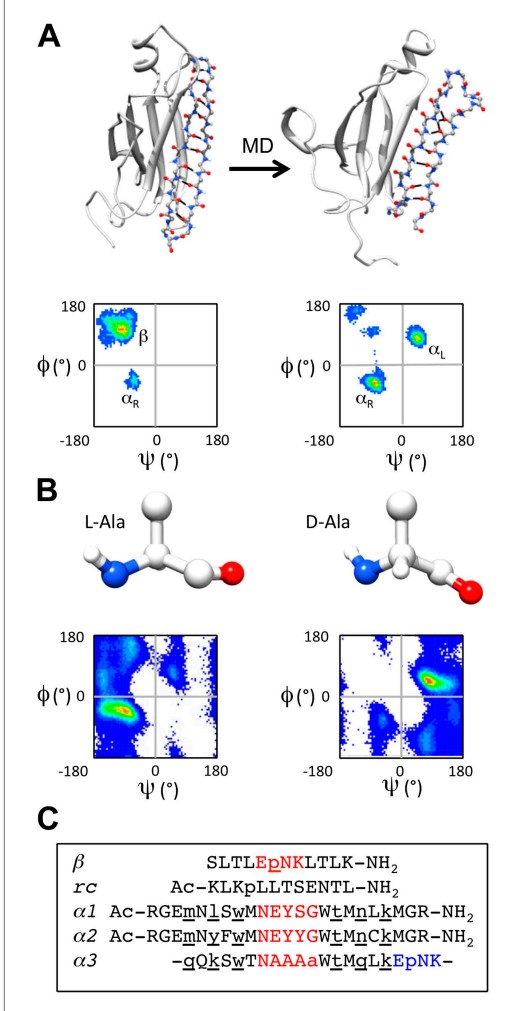

**Figure 1**. α-sheet conversion, conformational properties and peptide designs. (**A**) β- to α-sheet conversion of transthyretin (as reported by **Armen et al., 2004a**, **2004b**). The protein backbone is shown in cartoon representation with the region of interest (residues 105–121) shown as sticks. At 0 ns (top left) the residues of interest form a β-hairpin. The dihedral angles for 1 ns of dynamics of these residues are found mainly in the β-region of the Ramachandran plot (top left quadrant, lower left panel; increasing frequency of occupancy is shown from blue through red) with several turn residues in the $\alpha_R$ conformation (bottom left quadrant). After 30 ns (top right) the β-sheet has converted to an α-sheet. The dihedral angles for 1 ns of dynamics of the same residues reveal that the majority of residues have moved from the β-region to the $\alpha_L$ (top right quadrant) or $\alpha_R$ region of the Ramachandran plot (lower right panel). (**B**) Intrinsic residue propensities for L- and D-alanine were calculated from 100 ns of MD simulations of a GGXGG peptide system (**Beck et al., 2008**) (D-alanine was simulated using the same protocol). The backbone structure (upper panels) as well as the Ramachandran plot of the conformation of the alanine

alternate between the $\alpha_L$ and $\alpha_R$ conformations (**Daggett, 2006**) (**Figure 1A**). Although locally helical, the alternating dihedral angles form an extended chain resulting in the carbonyl groups and amide groups aligning in a plane. Such an arrangement creates uniform electrostatic faces to aid in the addition of further strands (**Armen et al., 2004b**). Once a sheet is formed, a simple peptide plane flip could convert the α-sheet into a β-sheet and ultimately a mature fibril (**Daggett, 2006**). α-sheet structure has been observed in a peptide crystal structure (**Di Blasio et al., 1994**), and short stretches of α-strand are present in various proteins in the Protein Data Bank (**Daggett, 2006**), although extensive α-sheet formation has not been observed in native proteins.

To investigate the role of α-sheet in amyloid formation, we computationally designed numerous small, stable α-hairpin peptides. We reasoned that if α-sheet is the novel backbone structure in toxic oligomers, then α−sheet peptides designed to be complementary to the structure in the oligomer should bind to the toxic oligomer and inhibit amyloidosis. Our designs began with a backbone template in an ideal α-sheet conformation. We then investigated combinations of residues with high propensities to populate desired regions of conformational space using our Structural Library of Intrinsic Residue Propensities (SLIRP), which is part of our Dynameomics project (**Beck et al., 2008**; **van der Kamp et al., 2010**). Owing to the expected transient nature of the $\alpha_L$ conformation, we stabilized the structure using D-amino acids, which essentially have inverted conformational propensities compared with their normal L-counterparts (**Figure 1B**). Peptides containing alternating D- and L-amino acids have previously been shown to form extended structures (**De Santis et al., 1974**; **Heitz et al., 1981**), and they populate similar conformational space as our MD-identified α-sheet sequences consisting solely of L-amino acids. MD simulations (**Beck et al., 2000–2014**) were performed to assess the stability of the de novo designed amino acid sequences. Sequences designed to adopt β-sheet and random coil conformations were included in our experiments as controls.

Several of the most promising peptide designs were selected for experimental characterization in two different amyloid systems: transthyretin (TTR) and beta amyloid 1-42 [Aβ(1−42) or Aβ for short]. Both are associated with amyloid diseases, systemic in the case of TTR and Alzheimer's Disease in the case of Aβ, but they have completely different sequences and structures. Aβ is a largely unstructured peptide fragment in aqueous

*Figure 1. Continued*
residue during the entire simulation, demonstrate the
conformational preference for L-alanine to adopt
the $\alpha_R$ conformation and for D-alanine to favor the
$\alpha_L$ conformation (lower panels). (**C**) Peptide designs
reported in this study. All designs are single turn
hairpins, with the exception of *α3*, which contains a
cyclic peptide backbone resulting in two turns. Hairpin
peptides are N- and C-terminally acetylated and
amidated, respectively, except for *β*, which had a free
N-terminus. D-amino acids are denoted by lower case
and are underlined, and turn residues are colored red in
the linear peptides and red and blue in the cyclic design.

solution, while TTR is a tetramer composed of immunoglobulin-like β-sandwich domains. Both systems have well characterized aggregation profiles and aggregate under mild, non-denaturing conditions (*Quintas et al., 1997*; *Foss et al., 2005*; *Hopping et al., 2013*). Here we focus on five peptide designs (*β, rc, α1, α2, α3*) (*Figure 1C*). *β* is the designed Trpzip 3 β–hairpin, but with Trp to Leu substitutions to improve spectroscopic properties (*Cochran et al., 2001a,b*). The β-design was included as a negative control for TTR and a positive control for Aβ. β-hairpins are known inhibitors of Aβ aggregation (*Yamin et al., 2009*; *Cheng et al., 2012*). The *rc* peptide was designed to be an unstructured random coil to provide a negative control; it is a randomly scrambled version of the β-sheet sequence, *β*. The remaining three peptides were designed to adopt α-sheet structure. *α1* and *α2* are linear hairpins containing a sheet of alternating D- and L-amino acids. *α1* was designed to have high α-sheet propensity. *α2* is a derivative of *α1* with modifications aimed to improve stability and the introduction of a Cys for coupling experiments. *α3* consists of a sheet of alternating D- and L-amino acids and two turns, creating a cyclic amide backbone. Despite our best efforts we were unable to design a soluble, random coil control based on the α-sheet peptides, i.e. the same composition and length. Shuffling of the amino acid sequences resulted in insoluble peptides that were unusable in the solution-phase assays, so we opted to use smaller but well-defined controls.

## Results and discussion

### Inhibitory properties of designs in two different amyloid systems

First we tested our peptides for anti-amyloidogenic activity in a fibrillization assay using transthyretin (TTR). Four of the five designed peptides (*α2* was sparingly soluble and therefore not tested in any solution-phase experiments) were co-incubated with TTR (in excess 20:1, expressed relative to TTR monomer) at pH 4.5 to trigger dissociation of the native tetramer followed by aggregation (*Quintas et al., 1997*; *Foss et al., 2005*). Note that a 10:1 ratio (and higher) is common in these aggregation inhibition assays (*Frydman-Marom et al., 2011*; *Hochdorffer et al., 2011*). The aggregation was monitored via binding of Congo red (*Figure 2A*). The percentage inhibition of aggregation was determined at 48 hr, after the aggregation stabilized. At these concentrations, *α1* and *α3* resulted in 72 and 56% inhibition, respectively, relative to TTR alone at low pH. In contrast, the *rc* and *β* controls resulted in little to no inhibition. The neutral pH tetrameric TTR control changed little over time. Moreover, the designs in the absence of TTR did not bind Congo red and were indistinguishable from the buffer-only controls (*Figure 2—figure supplement 1*). To ensure that the observed increase in Congo red binding reflected the formation of amyloid fibrils, we performed atomic force microscopy (AFM) (*Figure 2—figure supplement 2*).

Since our aim is to target the toxic oligomer, we determined when the toxic oligomeric species was present during the course of aggregation. The toxicity of TTR was assessed by monitoring cell viability using the MTT assay after treating SH-SY5Y neuroblastoma cells with TTR that had been allowed to aggregate for different periods of time at pH 4.5. Toxicity was apparent around 24 hr, and the results from this time point are shown in *Figure 2B*. Under these conditions, the viability of the treated cells was reduced by approximately 20%, indicating that TTR was aggregating via the toxic pathway. Addition of the controls, *rc* or *β*, to a 24-hr pre-incubated TTR sample had little to no effect on further aggregation, as detected with the Congo red assay. However, when *α1* and *α3* were mixed with 24-hr pre-incubated, toxic TTR, they inhibited 81% and 77%, respectively, of the remaining TTR aggregation observed in the absence of inhibitor.

Similarly, we co-incubated our designs (in 10:1 excess) with Aβ at pH 7.4 and followed fibril formation with thioflavin T (ThT) fluorescence changes upon binding (*Figure 2C*). The presence of fibrils was confirmed by AFM spectroscopy (*Figure 2—figure supplement 3*). As observed for TTR, both *α1* and *α3* inhibited the levels of fibril formed. Again *α1* was more efficient, inhibiting approximately 87%

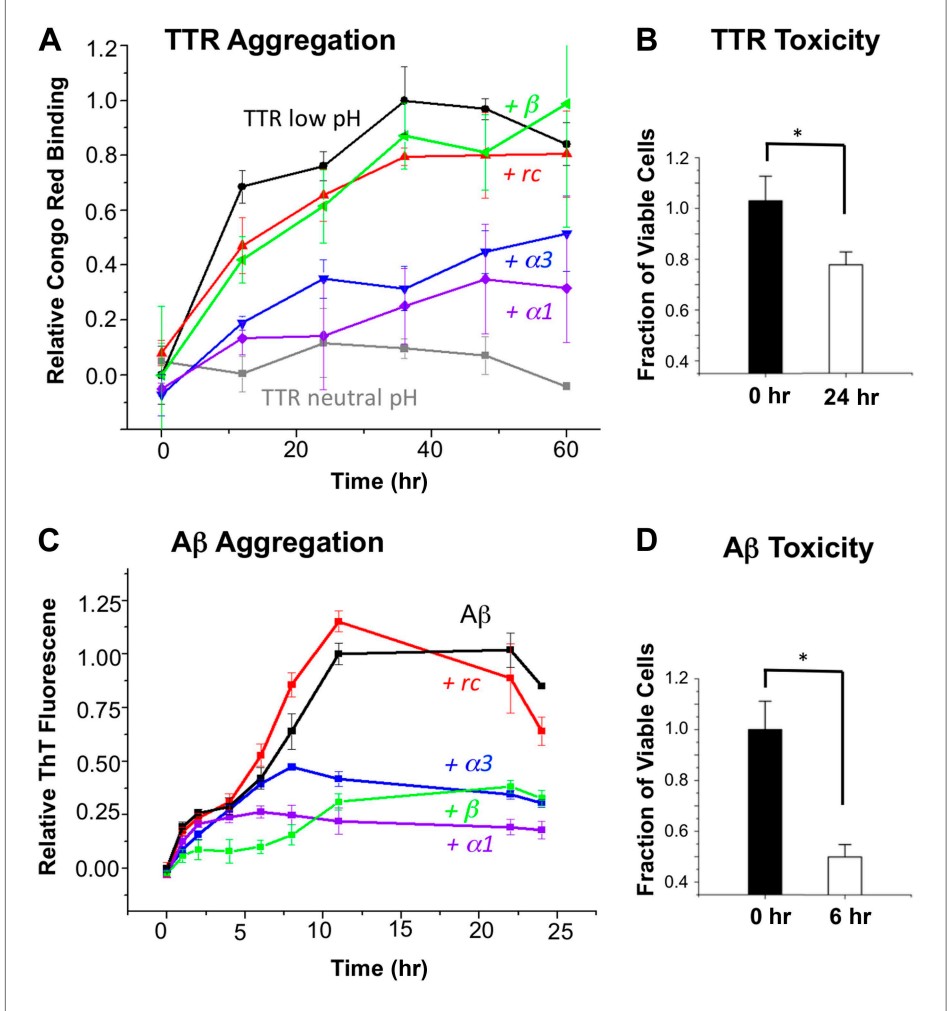

**Figure 2**. α-Sheet designs inhibit amyloid formation and selectively bind toxic species. (**A**) Peptide designs (800 μM) were co-incubated at pH 4.5 with 40 μM TTR (monomer) at 37°C and aggregation was monitored by Congo red binding. Error bars indicate the standard deviation. (**B**) Toxicity of the TTR solution after 24 hr pre-incubation at pH 4.5 against the human neuroblastoma cell line SH-SY5Y in a MTT metabolic viability assay. (**C**) ThT monitoring of 10 μM Aβ aggregation and inhibitory effects of 100 μM peptides present from the beginning of the aggregation at 37°C. Inhibition values were taken at 12 hr due the decay in ThT fluorescence, particularly for uninhibited samples, which has been described elsewhere (*Yamin et al., 2009*). (**D**) Aβ toxicity after 6 hr of aggregation, as probed using the MTT assay and the SH-SY5Y cell line. All data represent average ± SD (* indicates p<0.05, determined using Student's *t* test).

The following figure supplements are available for figure 2:

**Figure supplement 1**. Increase in Congo red binding is not due to peptide aggregation.

**Figure supplement 2**. AFM spectroscopy reveals aggregation conditions ultimately result in fibrils.

**Figure supplement 3**. Increase in ThT fluorescence is not due to peptide aggregation.

**Figure supplement 4**. AFM spectroscopy confirms an increase in fibrillar products.

of fibril formation compared with 66% for *α3*, when measured after the reaction stabilized at 22 hr. The β-sheet design, *β*, also resulted in 62% inhibition of Aβ fibrillization, which began exerting its effect primarily between 0 and 6 hr. In contrast, *α1* became inhibitory after aggregation proceeded

for approximately 6 hr. The peptide designs alone did not alter the fluorescence of ThT, and, as with Congo red, they were indistinguishable from buffer–only controls over the time course of the assay (*Figure 2—figure supplement 4*). Also note that the α-sheet designs are effective at lower concentrations and the efficacy increases at lower temperature; for example, a fourfold excess of *α1* essentially completely abolishes Aβ aggregation at 25°C (unpublished). We report the 37°C results here, though, for comparison with TTR, which aggregates more slowly at 25°C.

Aβ toxicity was also assessed using the MTT assay and found to reduce the viability of treated SH-SY5Y cells to less than 50% after 6 hr incubation (*Figure 2D*). Addition of the α-sheet designs to a pre-aggregated (6 hr), toxic sample of Aβ showed essentially complete inhibition of 97% for *α1* and 96% for *α3*, compared with the extent of remaining aggregation observed in the absence of inhibitor.

## Immobilization of designs

Despite much effort over the last few years it has not proved possible to isolate and characterize toxic soluble oligomers. So, to further probe which species our peptide designs are binding to, we immobilized the designs on agarose beads and applied solutions of either fresh or pre-incubated, toxic samples of Aβ and TTR. Immobilization in this manner also allowed us to test the sparingly soluble *α2* design by limiting self-aggregation. The peptides were immobilized via their lysine residues on aldehyde-functionalized agarose beads. Their ability to bind TTR or Aβ from solutions at various stages of aggregation was investigated using dot blot analysis of the eluents. All three immobilized α-sheet designs (*α1*, *α2* and *α3*) bound significantly more TTR from pre-incubated, toxic oligomer-containing TTR solutions (pre-incubated at low pH for 24 hr) than did the *rc* and *β* controls (*Figure 3A*). Note that while *α2* does not perform as well as the other two designs (*Figure 3A*), the binding of TTR by *α2* relative to the *β* and *rc* controls is significant, as shown by the statistical analysis in *Figure 3—figure supplement 1*. The extent of TTR binding by the *β* and *rc* controls is the same as that of the column matrix alone, which does bind some TTR in the absence of immobilized peptide (*Figure 3—figure supplement 2*). *α1*, *α2* and *α3* also bound pre-aggregated, toxic Aβ preferentially, whereas *β* and *rc* did not (*Figure 3B*, and see statistical analysis in *Figure 3—figure supplement 3*). In the case of the *β* design, it preferentially bound the fresh, monomeric Aβ solutions (0 hr) over the toxic solutions (6 hr) (*Figure 3C*), indicating that inhibition was due to interactions with the 'native' form, not the toxic oligomer. β-hairpins are known inhibitors of Aβ aggregation (*Yamin et al., 2009*). In contrast, α-sheet, as demonstrated with *α1*, did not bind native, tetrameric TTR nor fresh, monomeric Aβ but instead preferentially bound species from the toxic, aggregated 6 hr samples (*Figure 3D*). Moreover, the α-sheet designs did not bind the fibrillar forms of Aβ or TTR acquired by allowing the aggregation reactions to continue for over 3 weeks, as illustrated with immobilized *α1* (*Figure 3—figure supplement 4*).

## Biophysical characterization of designs

The instability of α-sheet structure in proteins and peptides containing solely L-amino acids leaves us with no established spectroscopic signatures with which to assess our structures. We can, however, make and test predictions based on the unique conformational and electronic environments resulting from this structure. Circular dichroism (CD) signals arise from the differential absorption of left- and right-hand polarized light by chiral molecules. For proteins, the orientation of individual amide bonds is responsible for the resulting CD spectra in the far-UV region. Mirror image structures, formed by replacing whole L-amino acid sequences with D-amino acids, such as gramicidin A, produce mirror-image CD-spectra (*Koeppe et al., 1992*). We anticipate that the α-sheet structure would be effectively invisible due to near equal absorbance of both left and right polarized light, with any residual signal emanating from the turns and terminal residues.

The electrostatic interactions between aligned amide groups in an α-sheet (*Armen et al., 2004a*; *Daggett, 2006*) are expected to give rise to strong Fourier-transform infrared (FTIR) signals. Torii recently performed density functional theory calculations of three slightly different orientations of α-sheet structures (*Torii, 2008*). All models featured a strong high-frequency absorbance in the 1675–1680 cm$^{-1}$ region, with a weaker band around 1640 cm$^{-1}$, which appears to be distinct from α-helix (~1650–1658 cm$^{-1}$), β-sheet (~1620–1640 cm$^{-1}$) and turn structures (~1670 cm$^{-1}$) (*Barth and Zscherp, 2002*).

Nuclear magnetic resonance (NMR) spectroscopy can provide site-specific conformational information. Owing to the unique alignment of the amide groups in an α-sheet, we expect to see strong sequential $d_{NN}$ Nuclear Overhauser Effect (NOE) crosspeaks along the backbone since the NH groups

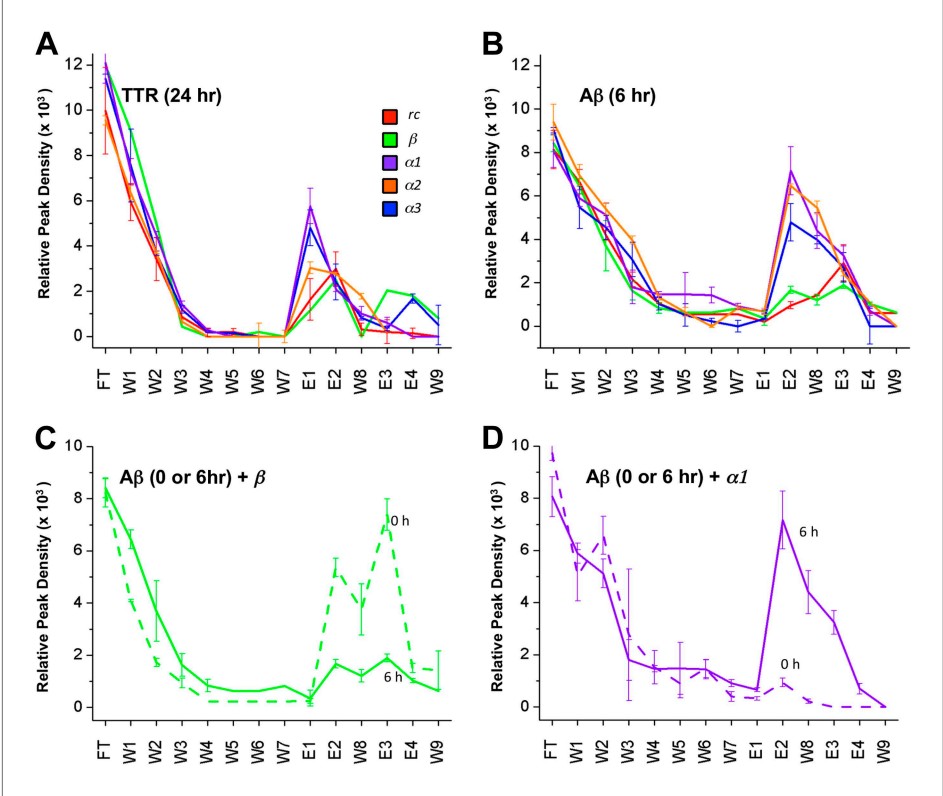

**Figure 3**. Immobilized designs bind toxic soluble oligomer from solution. Peptide designs were immobilized onto agarose beads and their ability to bind TTR or Aβ from solution at various stages of aggregation was probed using dot blot analysis. The presence of TTR or Aβ in the initial flow through (FT), sequential buffer washes (W), and sequential guanidine hydrochloride elution steps (E1–E2 and E3–E4) (x-axes) was detected by the integrated peak density of the dot blot analysis (y-axes). E1–W9 are within the linear range of the immunochemistry. (**A**) All three α-sheet designs, *α1, α2* and *α3* more strongly bound species from the 24 hr pre-aggregated, toxic TTR solutions than did either control. (**B**) Similar results were observed with the α-sheet designs binding to toxic Aβ solutions pre-aggregated for 6 hr. Despite the inhibitory effects seen with the *β* design in the Aβ fibrillization assay, little Aβ from a pre-aggregated solution bound to the immobilized *β* design. (**C**) Comparison of binding from a fresh (0 h), or pre-aggregated (6 hr), toxic Aβ solution. β is the only design that preferentially bound fresh Aβ over the aggregated toxic form, indicating that the inhibition observed was due to interactions with monomeric Aβ. (**D**) In contrast to the β control, α1 preferentially bound the pre-aggregated, toxic form of Aβ compared with fresh monomeric Aβ.

The following figure supplements are available for figure 3:

**Figure supplement 1**. Statistical analysis of data presented in *Figure 3A*.

**Figure supplement 2**. Nonspecific binding of nonnative TTR to column matrix.

**Figure supplement 3**. Statistical analysis of data presented in *Figure 3B*.

**Figure supplement 4**. Immobilized α-sheet designs do not bind fibrils.

are aligned on one side of the chain instead of alternating between opposite faces as in β-sheet structure. Furthermore, we would not expect to observe the long-range $d_{NN}$ or the strong sequential $d_{aN}$ NOEs characteristic of β-sheets. Thus, CD, FTIR and NMR studies were performed to assess the structures of our peptide designs.

The CD spectrum of *rc* shows a random coil signal with a strong negative absorption around 200 nm (*Figure 4A*). In its FTIR spectrum we observe multiple small absorbance peaks not corresponding to a predominance of any specific structure (*Figure 4B*). Our β-sheet control is a modified version of the

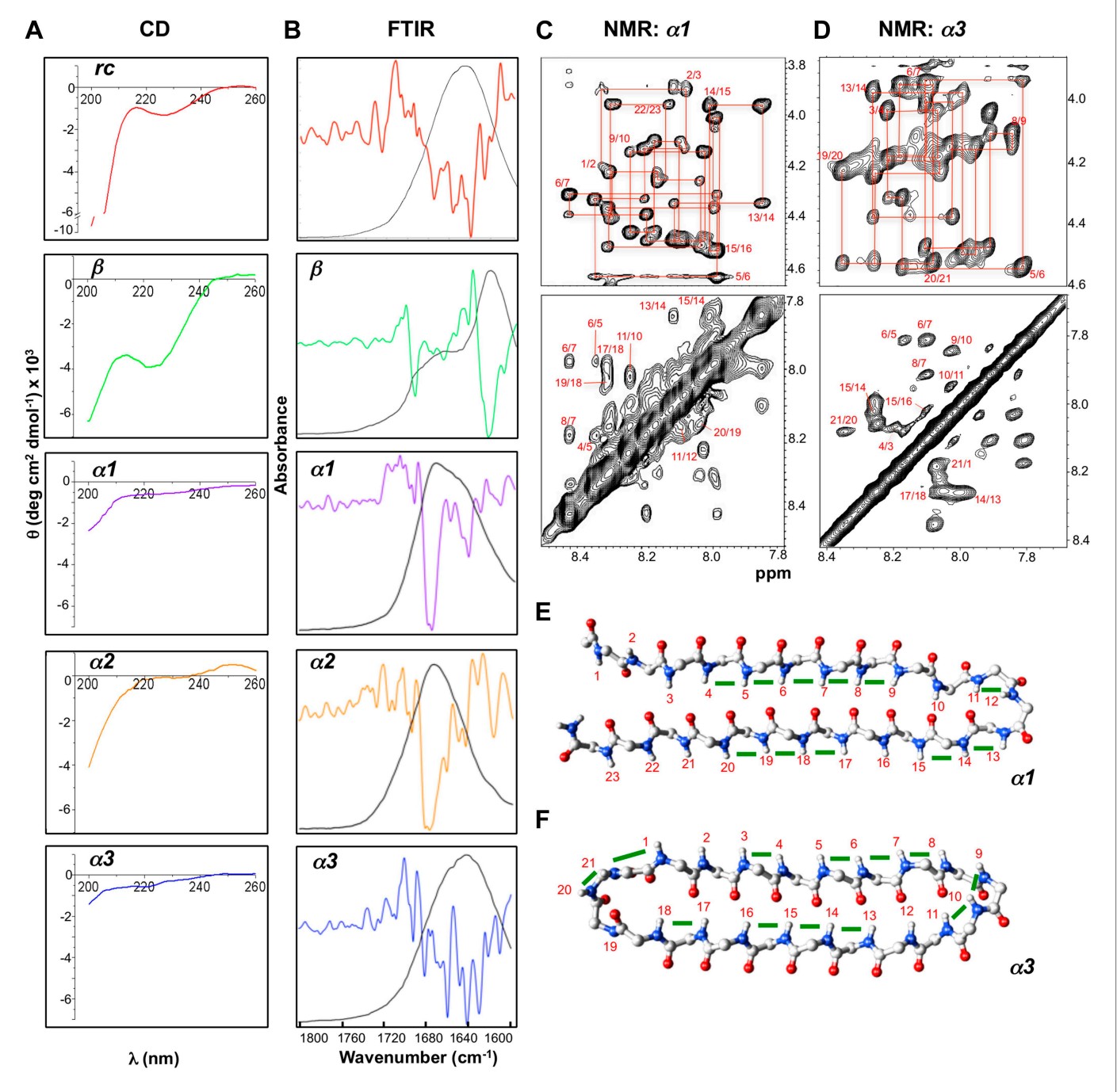

**Figure 4**. Designed peptides display unique spectroscopic signatures expected for α-sheet. (**A**) CD spectra for peptide designs reveal a random coil structure for *rc* and β-structure spectrum with a bit of random coil for *β*. In contrast, *α1*, *α2* and *α3* have largely featureless CD spectra with some random coil content expected to arise from the turns and tail residues. Note the different scales for the y-axes. All spectra are presented as molar ellipticity, highlighting the difference in intensity of the random coil component for each design compared with the *rc* spectrum. (**B**) FTIR spectra of the peptide designs, displayed as both absorbance (black line) and the second derivative (colored line), correlate well with the CD spectra. The *β* design shows a strong signal at 1632 cm$^{-1}$, as expected for β-structure. The α-sheet designs have signals near 1640 and 1675–1680 cm$^{-1}$ and the absorption is more intense for *α1* and *α2*. (**C**) Fingerprint (top) and NH region (bottom) of the $^1$H NOESY spectra for *α1*. Sequential assignments are shown in red and multiple sequential NOEs are observed and labeled. (**D**) Fingerprint (top) and NH region (bottom) of the $^1$H NOESY spectra for *α3*. The NH region reflects the predominance of NH–NH interactions and lack of other main-chain interactions characteristic of the common secondary structures. Mapping backbone NOEs on computational models as green bars (**E**, *α1* and **F**, *α3*) reveal in-plane alignment of the peptide groups along the majority of the

*Figure 4. Continued on next page*

*Figure 4. Continued*

sheet. NOEs in the turn regions determined whether the carbonyl or amide hydrogens pointed up in the structures as oriented in the figure (N-terminus top left). Cα, C, N, H and O atoms are shown in gray, gray, blue, white and red, respectively.

The following figure supplements are available for figure 4:

**Figure supplement 1**. Distances corresponding to $d_{NN}$ NOEs calculated from MD simulations.

**Figure supplement 2**. Distances corresponding to intraresidue $d_{\alpha N}$ NOEs calculated from MD simulations.

**Figure supplement 3**. Distances corresponding to sequential $d_{\alpha N\,(i-i+1)}$ NOEs calculated from MD simulations.

Trpzip peptide, which forms a stable β-hairpin in solution. In agreement with previous structural work (*Cochran et al., 2001a*), *β* exhibits a CD spectrum reflective of β-structure and random coil, with a minimum near 220 nm and another near 200 nm (*Figure 4A*). The substitution of Leu for Trp removed the strong exciton-coupling between the Trp residues observed in the parent peptide thereby 'exposing' the β-structure CD signal. The β-sheet structure was confirmed by FTIR through its strong absorbance at 1632 cm$^{-1}$ (*Figure 4B*). The CD spectra for *α1*, *α2* and *α3* are essentially featureless, as expected for the cancellation of $\alpha_L$ and $\alpha_R$ signals, except for a slight dip around 200 nm consistent with turn formation (*Figure 4A*). *α1* and *α2* exhibited the predicted FTIR α-sheet bands at 1640 and 1675–1680 cm$^{-1}$ (*Figure 4B*). These bands were less pronounced in *α3*. Cyclization of the amide backbone of *α3* through a non-optimal turn may have caused distortion of the structure, as has been reported in other peptide systems (*Clark et al., 2005*), and suggests an area for improvement in future designs. Altogether, these results prove that the designed α-sheet peptides do not form α-helix or β-sheet structure, and that the random coil content is not large (compare against the scale of the *rc* control CD spectrum). Thus, these results are consistent with and supportive of the designed α-sheet structure.

We performed further structural studies utilizing homonuclear NMR spectroscopy. Multiple sequential $d_{NN}$ NOEs were observed along the backbones of both *α1* and *α3* (*α2* was not soluble enough for NMR) (*Figure 4C,D*). No long-range $d_{NN}$ or $d_{\alpha N}$ NOEs indicative of α-helical or β-sheet structure (*Wüthrich, 1986*) were observed. The NH···NH NOEs are mapped onto structural models of *α1* and *α3* (*Figure 4E,F*), highlighting the stretches of α-strand structure in both designs. As mentioned above, strong sequential $d_{NN}$ NOEs are expected for α-sheet, but not for β-sheet, and strong sequential $d_{\alpha N}$ NOEs are expected for β-sheet, but not for α-sheet. To test this idea we calculated the ensemble-weighted NH–NH, intraresidue Hα-NH and sequential Hα-NH distances in MD simulations for *α1* in an α-sheet conformation and a β-sheet conformation, as well as a natural β-hairpin within a protein (*Figure 4—figure supplements 1–3*). The calculated NH–NH distances in the strands are much shorter in the α-sheet, consistent with the strong $d_{NN}$ NOEs observed for both *α1* and *α3*, and the observed breaks in the patterns (*Figure 4E,F*) reveal vulnerabilities in the structures and provide direction for improved designs. The bulk of the observed intraresidue and sequential $d_{\alpha N}$ NOE intensities in *Figure 4C,D* are of similar magnitude (12/21 for *α1*) or the intraresidue NOE is stronger (6/21 for *α1*), consistent with α-sheet structure (*Figure 4—figure supplements 2,3*). There are, however, three residues with strong sequential $d_{\alpha N}$ NOEs in *α1* but all three of these are involved in $d_{NN}$ NOEs, which is inconsistent with β-structure.

We observed no long-range side chain-side chain NOEs despite increasing the mixing time of the NOESY experiments up to 400 ms, perhaps due to residual dynamics in the peptide. The lack of these distance restraints prevented the generation of a well-converged solution structure; however, in support of the CD and FTIR spectroscopic data, the NMR data are consistent with α-sheet secondary structure and inconsistent with α-helix, random coil, and β-sheet structures.

## Conclusions

Ten years ago a common conformation was demonstrated among soluble oligomeric species from amyloid proteins/peptides of diverse sequence and structures that cross react with the A11 antibody (*Kayed et al., 2003*). Also at that time we identified a novel target structure, α-sheet, through MD simulations and proposed that it is the defining feature of the toxic oligomeric species (*Armen et al., 2004a*; *Daggett, 2006*). Unfortunately, the precise structure of this toxic intermediate remains elusive,

and it has become clear that the oligomers are conformationally heterogeneous (*Carulla et al., 2009*; *Bitan et al., 2005*, and references therein). Here we have taken a different approach to probe these soluble oligomeric species through experimental test of our α-sheet hypothesis through peptides designed to be complementary to the proposed α-sheet structure in the oligomeric intermediates. Three of these computationally derived designs were synthesized and characterized experimentally, and they do indeed appear to adopt α-sheet structure (as shown by FTIR, CD and NMR). The two soluble α-sheet designs (*α1* and *α3*) inhibited both TTR and Aβ aggregation in solution. In addition, when immobilized to agarose beads, all three α-sheet designs bound species from the toxic TTR and Aβ preparations preferentially over the nontoxic fresh samples. In contrast, the *β* control formed a β-hairpin, as supported by CD and FTIR, and it preferentially bound the monomeric form of Aβ and it did not react with TTR. When mature fibrils were applied to the immobilized α–sheet designs, the fibrils did not bind and appeared to have no affinity for the α-sheet structure.

While these α-sheet peptides were not designed against a specific protein target, we observed inhibition of aggregation in two very different amyloid systems. These results support our hypothesis that α-sheet structure is involved in the toxic oligomeric stage of aggregation, and they provide a reference to determine spectroscopic signatures that can now be used to investigate the structural changes amyloid proteins undergo during amyloidosis. In addition, the α-sheet designs may aid in capture of the elusive toxic oligomeric species for in-depth characterization. Having demonstrated that the α-sheet structure may constitute a broad-based inhibitor of amyloidosis, our α-sheet designs introduce a novel class of amyloid inhibitors that target the toxic soluble oligomeric state of different amyloidogenic peptides and proteins.

## Materials and methods

### Computational design

The α-sheet peptides were designed *in silico* using our database of amyloid protein MD simulations to determine preferred backbone geometries. Our SLIRP database (*Beck et al., 2008*; *van der Kamp et al., 2010*) was used to select residues with high propensity to adopt the desired structure. MD simulations were performed to assess both turn and α-sheet stability. Ideal α-sheet and β-sheet templates were created and sequences were chosen based on their intrinsic conformational preferences and to consist of a mix of polar and nonpolar amino acids to maintain good inter-strand interactions and solubility. Intrinsic conformational propensities of all 20 L-amino acids in a GGXGG peptide were determined by extensive molecular dynamics (MD) simulations (*Beck et al., 2008*). D-amino acid propensities were determined in a similar manner (manuscript in preparation). MD simulations were then performed to assess the stability of our designs. Multiple short simulations were performed, at least $3 \times 20$ ns, for each peptide at 25°C using our in-house MD package *in lucem* molecular mechanics (*il*mm) (*Beck et al., 2000–2014*), with the *Levitt et al. (1995)* all atom force field and the F3C water model (*Levitt et al., 1997*). Standard simulation protocols were followed (*Beck and Daggett, 2004*). α-Sheet stability was assessed by monitoring both the secondary structure and the turn structure, based on hydrogen bonding over the duration of the simulations. Results were expressed as the fraction of simulation time the atoms were within hydrogen bonding distance. Cα RMSD was used qualitatively to monitor backbone deviation from the ideal hairpin structure in conjunction with the hydrogen bond scoring function to determine promising designs. We took an iterative approach, with the results of the analyses used to modify and refine sequences for further simulation and evaluation. We chose several sequences from a pool of well-behaved simulations for experimental evaluation. High scoring designs were synthesized and their inhibitory effects were determined.

### TTR fibrillization assay

Aliqouts of transthyretin (TTR) (496-11; Lee Biosolutions, St. Louis MO) were made from a 1 mg/ml solution 20 mM ammonium carbonate, pH 8. Aliquots were lyophilized and stored at −18°C. Prior to use, TTR was dissolved to 80 μM (monomer) in acetate buffer (50 mM potassium acetate, 100 mM potassium chloride pH 4.5) and sonicated for 10 min. The stock solution was centrifuged before use. Peptide designs were added to stock TTR to a final TTR concentration of 40 μM (monomer) in acetate buffer (pH 4.5) in 500 μl microcentrifuge tubes., which were incubated at 37°C. Periodically, samples were collected from the TTR:peptide mixture by briefly centrifuging, and then carefully pipetting the solution up and down prior to withdrawing a 10 μl sample and diluting it in 190 μl of 10 μM Congo red

in an individual well of a 96-well assay plate. Each measurement was performed in triplicate. Absorbance measurements were taken at 477 and 540 nm. Relative Congo red binding was determined using the method of *Klunk et al. (1989)* via the following relationship: $rCb = (Abs_{540}/25,295)-(Abs_{477}/46,306)$. All datapoints were normalized to the value recorded for TTR alone pH 4.5 at 48 hr.

## Aβ fibrillization assay

Aβ(1-42) (AMYD-002; CPC Scientific, Sunnyvale CA) was stored as 2 mg/ml stock in hexafluoroisopropanol (HFIP) at −18 °C. Prior to use, the stock solution was thawed, an aliquot taken and the HFIP was removed under a gentle stream of air. A 1 mg/ml stock solution of Aβ was made in 5 mM NaOH and sonicated for 5–10 min. The stock was filtered through a 0.22 μm cellulose filter (Costar Spin-X; Corning Inc, NY). The concentration of stock Aβ was determined by first diluting the stock 1:50 in 5 mM NaOH then taking the absorbance at 220 nm ($\varepsilon_{220}$ = 50,000 cm$^{-1}$ M$^{-1}$). Aliquots of the NaOH stock were placed in separate wells of a 96-well black-walled fluorescence plate (Nunc) and immediately diluted in PBS (11 mM phosphate) containing 20 μM Thioflavin T (ThT) to give 150 μl of 10 μM Aβ at pH 7.5. Peptide inhibitors were added directly to 10 μM Aβ samples from concentrated aqueous stocks. Covered plates were incubated at 37°C and were periodically removed for fluorescence measurements. ThT fluorescence was measured at $\lambda_{ex}$ = 450 nm and $\lambda_{em}$ = 480 nm using a Tecan Safire2 plate reader.

## Immobilization and solution binding

Peptide designs were immobilized to the Pierce Amino Link resin following the manufacturer's instructions. Peptides were immobilized in a volume of 200 μl of coupling buffer (100 mM sodium phosphate, 150 mM sodium chloride, pH 7.2) and 2 μl cyanoborohydride solution (5 M sodium cyanoborohydride in 1 M NaOH) at a concentration of 358 μM overnight at 4°C. Any residual active sites were blocked with 400 μl quenching buffer (1 M tris hydrochloride, pH 7.4) and 4 μl cyanoborohydride solution for 4 hr at 25°C. Binding experiments were performed from 200 μl amyloid solution (5 μM Aβ or 20 μM TTR (monomer) diluted to the desired concentration in coupling buffer), which was allowed to bind to the peptide-bound agarose beads for 2 hr at 25°C. The solution was then collected by centrifugation (flow through, FT). The beads were resuspended in 300 μl coupling buffer, vortexed to obtain a uniform slurry, and the solution was again collected by centrifugation (wash 1, W1). The wash step was performed an additional five times (W2–W6). One final wash step was performed but after resuspending the resin, the solution was allowed to sit for 5 min before centrifugation (W7). The resin was next resuspended in 100 μl 2 M guanidine hydrochloride, incubated for 5 min at room temperature, then collected as before. This was performed twice (E1–E2). The resin was washed again with 300 μl coupling buffer (W8) followed by two elution steps with 6 M guanidine hydrochloride (E3–E4). One final wash step was performed with 300 μl coupling buffer (W9). All collected eluents were analyzed by applying triplicate 1 μl spots to nitrocellulose, and then performing standard dot blot analysis as described by Kayed et al. (9) with an anti-TTR (sc-13098, Santa Cruz Biotechnology, Santa Cruz, CA) or anti-Aβ (ab39377; Abcam Inc, Cambridge, MA) primary antibody diluted 1:1000 in 5% or 10% nonfat milk, respectively.

## SH-SY5Y cell viability

The toxicity of aggregates was tested against the human neuroblastoma cell line SH-SY5Y in an MTT cell viability assay. The human neuroblastoma cell line SH-SY5Y (CRL-226; American Type Culture Collection) was grown in 75 cm$^2$ flasks in 1:1 DMEM/F12 (CellGro, Manassas, VA) supplemented with 10% FBS and 50 units/ml penicillin/50 μg/ml streptomycin (complete media), and incubated at 37°C in humidified 5% CO$_2$ environment. Cells were routinely passaged when they reached 90% confluence by addition of trypsin (Gibco) and replated at a ratio of 1:10 in complete media. Cells were plated to a density of 25,000 cells/well in a 96-well plate (100 μl/well) and allowed to attach overnight. The cell assay was performed as described by *Reixach et al. (2004)*.

## CD spectroscopy

Far UV CD spectra were recorded on an Aviv model 420 spectrometer (Aviv Biomedical) over 200–260 nm in a 1 mm quartz cuvette. All samples were prepared at 100 μM, with the exception of the sparingly soluble design, *α2* which was prepared at 35 μM. All samples were prepared in 50 mM phosphate, 100 mM NaCl buffer, pH 5.8, and were recorded at 25°C with a resolution of 0.5 nm, a scan speed of 20 nm/min, and a 2 nm bandwidth. Average values from three scans were plotted using the Origin 8 software (Originlab, Northhampton, MA). All spectra were smoothed using the Savitzky-Golay method with 5–12 points/window, and polynomial order 2.

## FTIR spectroscopy

IR spectra were obtained using a Perkin–Elmer Spectrum 100 instrument equipped with a diamond attenuated total reflectance sample unit and an MCT detector. Peptide samples were pelleted and re-suspended as a 1–2 µl slurry. The slurry was applied to the diamond and dried to a film over a few minutes while following the disappearance of the broad liquid water band at ~1636 cm$^{-1}$ and the appearance of the protein amide I and amide II bands. The spectra were background-subtracted and comprised of 64 accumulations (4 cm$^{-1}$ resolution; 1 cm/s OPD velocity; strong apodization). Spectra shown here were recorded as soon as successively collected spectra (each recorded over 80 s) stabilized, indicating little further evaporation of liquid water. This approach was taken to eliminate spectral contributions from free liquid water without desiccating the peptide film any more than necessary. Second derivative spectra were calculated using the instrument software and 13 data points.

## NMR spectroscopy

Peptides were prepared in 50 mM potassium phosphate buffer containing 100 mM KCl pH 5.8. All NMR experiments were performed on Bruker Avance 600 and/or 500 MHz spectrometers equipped with cryogenic triple resonance probes. The sample temperature was kept at 25°C. 4,4-dimethyl-4-silapentane-1-sulfonic acid (DSS) was used for proton chemical shift referencing whereas indirect referencing was used for carbon and nitrogen. The resonance assignments for peptides were carried out using $^1$H-$^1$H TOCSY and [$^1$H-$^1$H] NOESY spectra recorded in 90% $H_2O$ and 10% $^2H_2O$. The assignments thus obtained are translated onto the natural abundance $^1$H-$^{15}$N HSQC and $^1$H-$^{13}$C HSQC spectra. All spectra were processed with Topspin3.0 (Bruker) and analyzed using CARA (http://cara.nmr.ch/doku.php) or Sparky (http://www.cgl.ucsf.edu/home/sparky/); figures were made using CARA.

## AFM spectroscopy

AFM was performed with a Dimension 3100 atomic force microscope using tapping mode and silicon tips (FESP; Bruker; Camarillo, CA). 10 µl of 10 µM Aβ(1-42) or 40 µM TTR were applied directly to freshly cleaved mica and incubated for 10 min. 50 µl of water was added then removed by capillary action with a lint-free lab wipe. A further 50 µl of water was added and incubated for 5 min before removal. The mica surface was then allowed to dry under ambient conditions prior to imaging.

## Acknowledgements

We thank Michelle McCully for excellent technical assistance during the course of this work and Dr Roger Armen for the initial studies that made this work possible. We also thank Drs Albert La Spada and George Martin for the use of laboratory space during the early stages of this work, and Drs David Baker, Pat Stayton, Ceci Giachelli and Buddy Ratner for generous use of equipment. Part of this work was conducted at the University of Washington NanoTech User Facility, a member of the NSF National Nanotechnology Infrastructure Network (NNIN).

## Additional information

### Competing interests

VD: Reviewing editor, *eLife*. The other authors declare that no competing interests exist.

### Funding

| Funder | Grant reference number | Author |
| --- | --- | --- |
| National Institutes of Health | GM-095808 | Valerie Daggett |
| National Science Foundation | CBET-0966977 | Valerie Daggett |
| Wallace H Coulter Foundation | | Valerie Daggett |
| Coins for Alzheimer's Research Trust | | Valerie Daggett |
| National Institutes of Health | GM-064440 | Gabriele Varani |
| National Institutes of Health | AI-074661 | James Bryers |
| National Institutes of Health | Intramural Research Program | Byron Caughey |

The funders had no role in study design, data collection and interpretation, or the decision to submit the work for publication.

## Author contributions

GH, Conception and design, Designed research, Performed research, Analyzed results, Wrote the paper; JK, Acquisition of data, Analysis and interpretation of data; RPB, Acquisition of data, Contributed unpublished essential data or reagents; PL, Conception and design, Contributed unpublished essential data or reagents; JB, GV, Analysis and interpretation of data, Drafting or revising the article, Contributed unpublished essential data or reagents; BC, Acquisition of data, Analysis and interpretation of data, Drafting or revising the article, Contributed unpublished essential data or reagents; VD, Conception and design, Analysis and interpretation of data, Drafting or revising the article, Contributed unpublished essential data or reagents

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
