## [Decision Letter]

Thank you for sending your work entitled “Designed α-sheet peptides inhibit amyloid formation by targeting toxic oligomers” for consideration at *eLife*. Your article has been favorably evaluated by a Senior editor, John Kuriyan, and 3 reviewers, one of whom is a member of our Board of Reviewing Editors.

The Reviewing editor and the other reviewers discussed their comments before we reached this decision, and the Reviewing editor has assembled the following comments to help you prepare a revised submission.

All reviewers agreed that your findings that peptides with a designed α-sheet conformation are capable of inhibiting amyloid formation by targeting toxic oligomers is a very important step towards characterizing the structural nature of pre-fibrillar, oligomeric states of proteins. All reviewers recommend publication after addressing some concerns that are summarized below:

Some concerns raised focus on the behaviour of the designed peptides.

1) Why don't the α-sheet peptides oligomerize themselves instead of binding to the sequence diverse targets, in particular at 20:1 design/target concentration ratios?

2) What is the net charge of designed peptides vs the net charge of the target? Can it be ruled out that the inhibition is not simply by electrostatic coating of the growing fibrils?

3) Is the predicted burial of the solvent accessible surface area smaller for α sheets, thus allowing the use of urea to bias away from competing β-sheet structure?

Other concerns focused on the interaction of the designed peptides with the oligomers.

4) Figure 3. The binding of TTR to the *rc* and the β peptides is explained by the background binding to the column material. However, the elution profile of the control experiment shown in Figure 3—figure supplement 2 looks different. These differences in profiles should be explained.

5) Figure 2. It seems that incubation with the *rc* peptide leads to a significant decrease in the aggregation of Aβ peptide. How can this be explained? The figure legend should be extended to better describe these results.

Finally, concerns were raised regarding the NMR characterization of the α-sheet peptides

6) The NMR data show that the peptides do not adopt an α-helical fold since typical sequential NOEs are missing. However, Figure 4 seems to indicate that the sequential alpha-amide NOE is stronger than the intraresidual one, which is expected for β-sheets. β-sheets can also have weak amide proton – amide proton NOEs. The authors should compare the experimental NOE intensities to those calculated from the simulations, in a population-weighted sense. Since they have the trajectories already, this should be possible with relatively little additional work. They should also compare the expected NOE patterns derived from the simulations of α-sheets and β-sheets and provide a table with the expected distances in α-sheets β-sheets.

7) In addition, the assignments seem to include some inconsistencies. For example, in Figure 4, bottom panel, the cross-peaks of 17/18 and 18/19 (which ought to be labeled 19/18) do not line up along the vertical dimension. This could possibly indicate that there is some heterogeneity in the peptide conformations. Similarly, several other cross-peaks have 'shoulders'. Is there any evidence for some heterogeneity or can this be explained differently?

---

## [Author Response]

*Some concerns raised focus on the behaviour of the designed peptides*.

*1) Why don't the* α*-sheet peptides oligomerize themselves instead of binding to the sequence diverse targets, in particular at 20:1 design/target concentration ratios?*

Our peptides designs are highly polar and somewhat amphipathic, with one completely polar face and the other mixed polar/nonpolar. They were designed to prevent self-oligomerization and we were successful. The alpha-sheet designs presented here do not form oligomers even at the very high concentrations used for the NMR studies. There is enough hydrophobic surface on the designs, to allow for binding to more hydrophobic amyloidogenic species, however. While not described here, we have some newer designs that are more hydrophobic and they do aggregate and some form fibrils. But, our goal is soluble non-aggregating designs, as presented here.

2) What is the net charge of designed peptides vs the net charge of the target? Can it be ruled out that the inhibition is not simply by electrostatic coating of the growing fibrils?

If electrostatic coating were the mechanism by which our designs inhibited aggregation we would expect α*3* (net charge +2) to be a better inhibitor than α*1* (+1), and it is not. Similarly, *β* (+2) does not inhibit TTR aggregation and *rc* (+1) doesn’t inhibit aggregation in either system, despite having the same charges as α*3* and α*1,* respectively. We also show that structure determines inhibition and that inhibition occurs during the aggregation process. These results are not consistent with a nonspecific electrostatic coating mechanism.

3) Is the predicted burial of the solvent accessible surface area smaller for α sheets, thus allowing the use of urea to bias away from competing β sheet structure?

*Other concerns focused on the interaction of the designed peptides with the oligomers*.

We do not anticipate any appreciable difference in solvent-accessible surface area between α-sheet peptides and β-sheets. We have demonstrated that the crankshaft motion to switch from α- to β-sheet can occur with little change in side-chain orientation(Armen et al. 2004; [18]), and therefore little change in SASA. Our design criteria were specifically chosen to prevent such complications by incorporating alternating D- and L-amino acid motifs. We are unsure what the reviewer means by using urea to bias away from competing β-sheet structure; we don’t perform any experiments with urea.

*4)*
Figure 3*. The binding of TTR to the* rc *and the β peptides is explained by the background binding to the column material. However, the elution profile of the control experiment shown in*
Figure 3—figure supplement 2
*looks different. These differences in profiles should be explained*.

The elution profile in the control experiment does look different because the matrix is different. *rc* and *β* both have peptide immobilized on the agarose, whereas the “blank” experiment is agarose blocked with tris. The blank surface is therefore very hydrophilic when compared to a surface coated with the peptides. It is unsurprising that TTR appears to elute more quickly from the polar blank column than it does from either column with immobilized peptides containing both hydrophobic and hydrophilic residues, as TTR is known to increase in hydrophobicity during aggregation, and therefore is expected to be retained longer under these conditions. We have added a short discussion of this in the legend to Figure 3—figure supplement 2.

*5)*
Figure 2*. It seems that incubation with the* rc *peptide leads to a significant decrease in the aggregation of Aβ peptide. How can this be explained? The figure legend should be extended to better describe these results*.

Finally, concerns were raised regarding the NMR characterization of the α-sheet peptides

The decrease in ThT fluorescence observed for the rc peptide is not necessarily a decrease in the amount of amyloid formed. Many other research groups have observed this phenomenon previously. For example, Anh and co-workers suggested that in their α-synuclein and Aβ experiments that the decrease in signal may be a result of unbound ThT quenching the binding fluorescence (Ahn et al. 2007). A similar observation was also reported by another group working on Aβ (47). There also appears to be a concentration dependence of this observation (Ahn et al. 2007), which could explain why we see a larger decrease in fluorescence for uninhibited Aβ (rc sample) than we do for our inhibited samples. This information has been added to the legend of Figure 2.

*6) The NMR data show that the peptides do not adopt an* α*-helical fold since typical sequential NOEs are missing. However,*
Figure 4
*seems to indicate that the sequential alpha-amide NOE is stronger than the intraresidual one, which is expected for β-sheets. β-sheets can also have weak amide proton – amide proton NOEs. The authors should compare the experimental NOE intensities to those calculated from the simulations, in a population-weighted sense. Since they have the trajectories already, this should be possible with relatively little additional work. They should also compare the expected NOE patterns derived from the simulations of* α*-sheets and β-sheets and provide a table with the expected distances in* α*-sheets β-sheets*.

To address this point the 1/r^6^ values were calculated for NH-NH pairs over the MD-generated ensembles, which should be proportional to the NOE intensity, and they are reported in distances as requested in a new figure: Figure 4—figure supplement 1. α*1* simulations were started from an α-sheet conformation and also from a β-sheet conformation, and presented as the average of triplicate 20 ns simulations at 298 K. We compared these results to MD of an antiparallel β-hairpin within a protein (PDB:1QAU). We found that in general the calculated intensities were stronger in the highlighted sheet regions in the alpha-sheet conformation, which is supportive of an α-sheet conformation rather than a β-sheet conformation. The tabulated data have been entered as Figure 4—figure supplement 1.

*7) In addition, the assignments seem to include some inconsistencies. For example, in*
Figure 4*, bottom panel, the cross-peaks of 17/18 and 18/19 (which ought to be labeled 19/18) do not line up along the vertical dimension. This could possibly indicate that there is some heterogeneity in the peptide conformations. Similarly, several other cross-peaks have 'shoulders'. Is there any evidence for some heterogeneity or can this be explained differently?*

We assigned the major conformation observed from the NMR for this work. There does appear to be some alternate, less populated conformations, but assignment was not possible. This is unsurprising for a small hairpin peptide lacking disulfide bonds or other covalent restraints. The flipped labeling of the crosspeak was corrected in Figure 4.